# Boosted high-throughput D⁺ transfer from D₂O to unsaturated bonds via Pd$^{\delta+}$ cathode for solvent-free deuteration

Xiu-Feng Zhang, Shi-Nan Zhang ⓞ, Zhao Zhang, Bing-Liang Leng ⓞ, Kai-Yuan Lu, Jie-Sheng Chen ⓞ & Xin-Hao Li ⓞ ✉

Deuterated organic compounds have gained significant attention due to their diverse applications, including reaction mechanism studies, probes for metabolism and pharmacokinetics, and raw materials for labeled compounds and polymers. Conventional reductive deuteration methods are limited by the high cost of deuterium sources (e.g., D₂ gas) and challenges in product separation and D₂O recycling. Electrochemical deuteration using D₂O is promising, but existing methods still suffer from low Faradaic efficiency (FE) and high separation costs. Herein, we report a deuterium ion diffusion-based all-solid electrolyser, featuring a RuO₂ anode for D⁺ generation from pure D₂O and a Pd/nitrogen-doped carbon-based liquid diffusion cathode (Pd$^{\delta+}$/NC LDC) with tunable electron deficiencies Pd$^{\delta+}$/NC to enhance selective deuteration. This system achieves over 99% selectivity for deuterated benzyl alcohol with a FE of 72%, and demonstrates broad applicability for the deuteration of aldehydes, ketones, imines, and alkenes with high FE and selectivity. Moreover, the Pd$^{\delta+}$/NC-based electrolyser can achieve ten-gram-scale production of deuterated benzyl alcohol over 500 hours, showcasing its potential for high-throughput, solvent-free deuteration reactions in practical applications.

Due to their wide range of applications, deuterated organic compounds have recently garnered significant attention; their applications include tools for reaction mechanism studies[1,2], probes for metabolism[3] and pharmacokinetics[4–6], and raw materials for labeled compounds and polymers[7,8]. The development of efficient reductive deuteration methods of various unsaturated bonds for deuterated compound production is always a critical priority to decrease the consumption of expensive D₂ gas or other deuterium sources (e.g., NaBD₄, DCl, and LiAlD₄) in conventional direct reduction methods[9–12]. The electrochemical deuteration method using relatively inexpensive deuterated water is very promising for the production of various deuterated compounds[13–17]; however, current strategies (Fig.S1a) still suffer from high separation costs of target products and difficulty in recycling D₂O from various solvents/electrolytes. Pioneering work

using an active palladium membrane has successfully reduced D₂O to D atoms on one side (Fig. S1b), and the substrate was reduced on the other side by the D atoms diffusing through the palladium membrane[18]; however, the diffusion efficiency of the D atoms through the palladium membrane and the total Faradaic efficiency (FE: ≤7% for deuterated benzyl alcohol) are still far from satisfactory for the mass production of various deuterated compounds. The exploration of efficient deuteration methods that enable high-throughput deuteration of various unsaturated compounds with satisfactory FE values and easy separation of D₂O for reuse with zero/low waste emission is still highly important.

Herein, we present the rational design of a deuterium ion diffusion-based all-solid electrolyzer with a pure substrate on the cathode side and a pure deuterium source D₂O on the anode side for

State Key Laboratory of Synergistic Chem-Bio Synthesis, School of Chemistry and Chemical Engineering, Shanghai Jiao Tong University, 200240 Shanghai, P. R. China. ✉e-mail: xinhaoli@sjtu.edu.cn

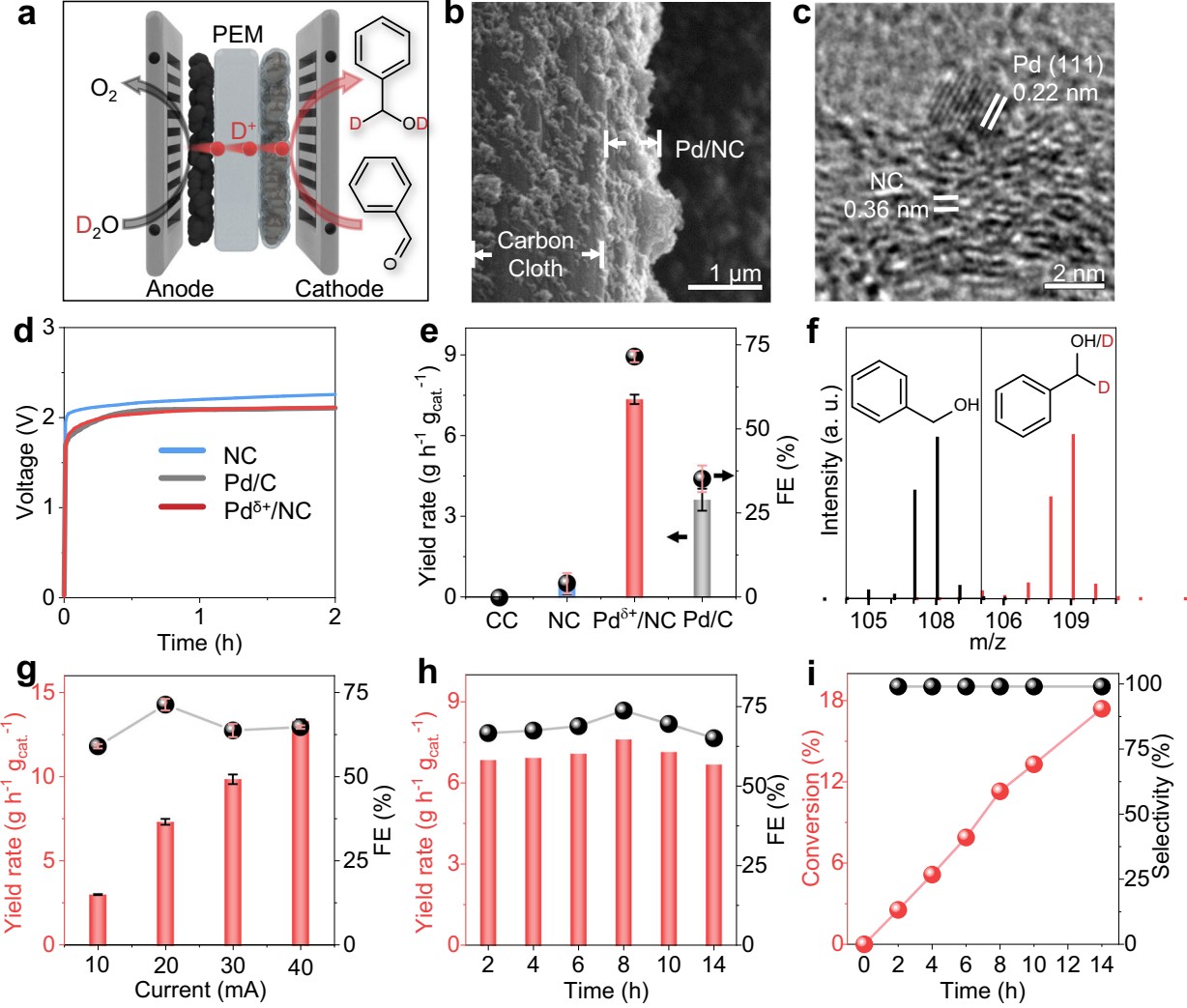

**Fig. 1 | D⁺ transfer from D₂O to benzaldehyde for solvent-free deuteration reactions. a** Schematic illustration of the deuterated reduction of benzaldehyde. **b** Scanning electron microscopy (SEM) image of Pd^δ+/NC LDC. **c** Transmission electron microscopy (TEM) image of Pd^δ+/NC. **d** Chronopotentiometry curves and (**e**) corresponding yields and FEs of deuterated benzyl alcohol from NC, Pd/C and Pd^δ+/NC. The typical test conditions are as follows: the anode catalyst is RuO₂ (2.1 mg cm⁻¹), 2 mL of pure benzaldehyde is circulated through the cathode, pure D₂O is circulated through the anode as the deuterium source, and galvanostatic

measurements are performed at 5 mA cm⁻². (The Ohmic resistance of the cell of Pd^δ+/NC and Pd/C are $0.45 \pm 0.01\,\Omega$ and $0.56 \pm 0.01\,\Omega$, respectively). **f** Mass spectra of benzyl alcohol and deuterated benzyl alcohol from the deuterated reduction of benzaldehyde. **g** Yield and FEs of Pd^δ+/NC LDC at different currents. **h** Yield and FEs of deuterated benzyl alcohol and (**i**) conversion and selectivity of benzaldehyde during long-term testing. Source data for Fig. 1d–i are provided as a Source Data file.

efficient solvent-free deuteration reactions. This electrolyzer is composed of a bench-marked RuO₂ anode as the deuterium ion generator and a Pd/nitrogen-doped carbon-based liquid diffusion cathode with tuneable electron deficiencies (Pd^δ+/NC) for facilitating selective deuterium ion generation and transfer to unsaturated substrates on the cathode; moreover, this electrolyzer could provide a selectivity exceeding 99% for deuterated benzyl alcohol production with a record-high FE of 72%. Furthermore, when pure D₂O is used as the deuterium source, the strategy demonstrates the universal production of C(sp³)−D-containing compounds from various aldehydes (FEs: 43–72%, selectivity: 88–99%), ketones (FEs: 31–90%, selectivity: 99%), imines (FEs: 11–40%, selectivity: 99%) and alkenes (FEs: 55–97%, selectivity: 99%). Under optimized conditions, the Pd^δ+/NC-based electrolyzer could achieve ten-gram-scale production of deuterated benzyl alcohol at an ampere-level current density for more than 90 h and could function even longer for 500 h at 20 mA; therefore, this electrolyzer has great potential of the high-throughput deuteration technique for practical use.

## Results

### Non-contact reductive deuteration via Pd^δ+ cathode

To achieve the contactless deuteration reaction between deuterium water and a pure unsaturated substrate, we designed an all-solid reactor (Fig. 1a and S2) to trigger the efficient transfer of D⁺ ions from D₂O oxidation on the anode via a proton exchange membrane (PEM) to the cathode without the assistance of any solvent or support electrolyte. Considering the well-developed OER catalysts (commercial RuO₂) and the membrane electrode assembly (MEA) process[19,20], we focus on the rational design of the cathode electrocatalysts to ensure the selective deuteration of specific substrates with satisfactory FE and current density for practical use. Pioneering work has shown that Pd metal is the benchmark active site for the deuteration of various unsaturated compounds using conventional direct reduction methods[21–23]. Thus, we fabricated a rationally designed liquid diffusion cathode (LDC) by coating a homogeneous layer of a well-designed Pd^δ+/NC on carbon cloth (Fig. 1b and S3). Both the particle size (Figs. S4–5) and the crystallinity (Fig. 1c and S6) of the as-embedded

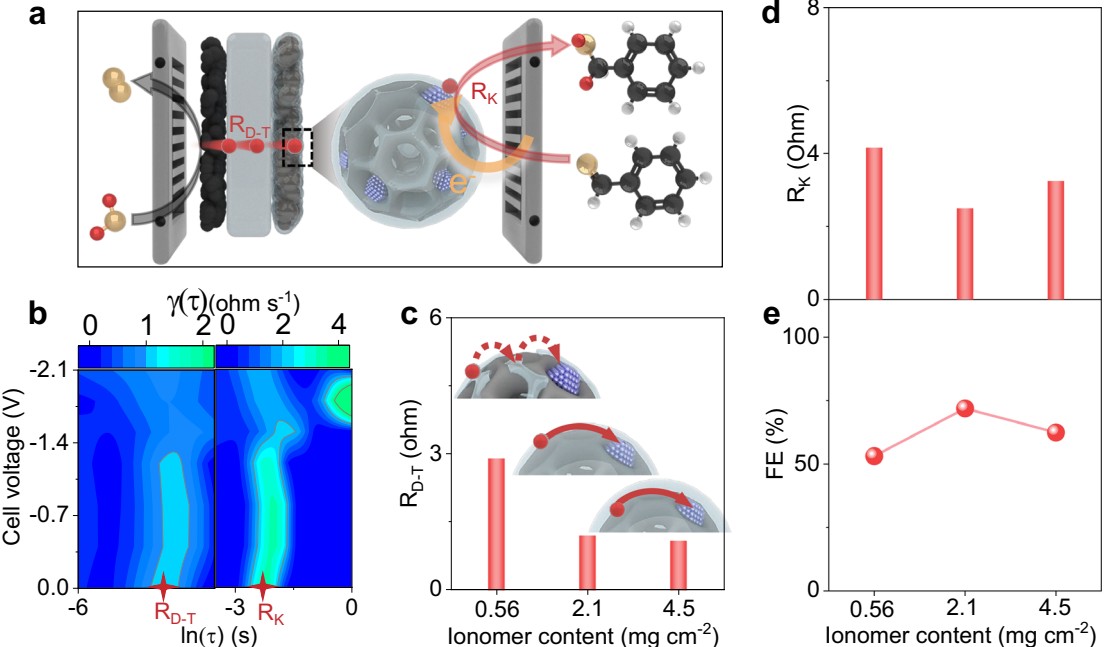

**Fig. 2 | Enhanced D⁺ transfer and kinetics of deuterated reduction by Pd^{δ+}/NC LDC.** **a** Schematic illustration of the deuterated reduction of benzaldehyde (blue, Pd; black, C; orange, O; red, D; white, H). **b** Distribution of relaxation times (DRT) results of Pd^{δ+}/NC LDC with different cell voltages. **c** D⁺ transfer resistance ($R_{D-T}$) of Pd^{δ+}/NC LDC with different ionomer contents (blue, Pd; red, D). **d** Resistance of reaction kinetics ($R_K$) and (**e**) FE of Pd^{δ+}/NC LDC with different ionomer contents. Source data for Fig. 2b–e are provided as a Source Data file.

Pd nanocrystals were effectively maintained during the process of cathode fabrication. The combination of the RuO₂-based MEA with the Pd^{δ+}/NC-based cathode produced a 3D network for possible D⁺ diffusion (red path in Fig. 1a) even without the involvement of any electrolyte.

Indeed, this all-solid reactor could trigger the solvent-free deuteration of benzaldehyde smoothly as well presented by the stable chronopotentiometry curves (Fig. 1d). Bare carbon cloth (CC), as the LDC, could not yield a detectable amount of deuterated benzyl alcohol (Fig. 1e). Notably, the Pd^{δ+}/NC LDC could trigger the continuous conversion of benzaldehyde to deuterated benzyl alcohol, as shown in the inset of Fig. 1a; this result was further confirmed by the mass spectra (Fig. 1f) and ¹H-NMR spectra (Fig. S7) of the as-formed product, with a yield rate of 7.39 g h⁻¹ g_{cat.}⁻¹ at 20 mA. The bare NC support with heteroatom dopants has an open band gap and thus the defect sites for possible activation of substrates. The NC-based cathode exhibits a detectable conversion of 47 μmol benzaldehyde (Fig. 1e) with a FE of 6%, which is only about one-tenth of Pd/NC cathode and one-fifth of Pd/C cathode. In addition, the Pd/NC cathode exhibited a wide electrochemical window for deuterated benzyl alcohol production, with FE values > 59% at different current densities ranging from 10 to 40 mA (Fig. 1g). Slightly increase in the current density from 10 to 20 mA leads to an elevated FE value from 59% to 72%. Further amplifying the current density to 40 mA may generate more hydrogen gas bubbles, resulting in a slight decrease in the FE values to 63–65% presumably due to the relatively sluggish substrate diffusion efficiency at the three-phase interface over the cathode. The Pd^{δ+}/NC LDC could achieve continuous conversion of benzaldehyde for more than 14 h (Fig. 1h–i) with notable selectivity for deuterated benzyl alcohol above 99% (black spheres in Fig. 1i), yield rates (bars in Fig. 1h) and FE values (black spheres in Fig. 1h). Under optimized conditions, the Pd^{δ+}/NC LDC could provide an FE of 72%; this value is nearly 1.8 times greater than that of Pd/C, 11 times greater than that of bare NC (Fig. 1e) and 10 times greater than that of the state-of-the-art Pd membrane electrode[18]. Thus, these results show the key role of Pd^{δ+} in improving the selective deuteration reaction.

**Reaction mechanism of reductive deuteration via Pd^{δ+} cathode**
Specifically, the Pd^{δ+}/NC LDC dyad enhances the D⁺ transport using a three-dimensional (3D) conducting network and the deuterated kinetics of benzaldehyde (Fig. 2a) according to the distribution of relaxation time (DRT) analysis results (Fig. 2b and S8–10)[24–26]. Increasing the ionomer content from 0.56 to 2.1 mg cm⁻² with fixed Pd loading (0.03 mg cm⁻²) clearly decreased the D⁺ transfer resistance ($R_{D-T}$) (Fig. 2b left) from 2.9 Ohm to 1.2 Ohm (Fig. 2c) and kinetic resistance ($R_K$) (Fig. 2b right) from 4.2 Ohm to 2.5 Ohm (Fig. 2d), respectively. Additionally, the further increase in the ionomer content to 4.5 mg cm⁻² did not further depress the $R_{D-T}$ but led to an undesired increase in the $R_K$, presumably due to the poison effect of the ionomer covering the Pd centers[27–30]; these results demonstrated an optimized ionomer content of 2.1 mg cm⁻², and this content was used for all following tests. Moreover, the $R_{D-T}$ and $R_K$ values of Pd^{δ+}/NC LDC were only half of those of Pd/NC LDC (Fig. S11). All these results indicated the key role of Pd^{δ+}/NC LDC with lower kinetic resistance in facilitating the deuteration reactions.

The synergy of the Pd/NC dyad of Pd^{δ+}/NC LDC in enhancing the deuteration of benzaldehyde was directly observed using in situ attenuated total reflectance surface-enhanced infrared absorption spectroscopy (ATR-SEIRAS). The key role of Pd^{δ+}/NC LDC as the active cathode in triggering the selective addition of D⁺ to benzaldehyde is shown by the following (Fig. 3a and Fig. S12): the decrease in the peak of −C = O of benzaldehyde at 1704 cm⁻¹ with increasing working voltage from −0.5 to −1.5 V[19]; and the gradual increase in the signals of the −O−D group at 2502 cm⁻¹ and the −C−O group at 1054 cm⁻¹ for deuterated benzyl alcohol over the Pd^{δ+}/NC LDC[31]. The density functional theory (DFT) simulation results further suggested a more favorable pathway for the reduction of D⁺ on the surface of the NC-supported Pd metal models (Fig. 3b and S13–14); here, compared to those of the deuterium evolution reaction (gray line) and the reductive addition to carbonyl oxygen (orange line), a much lower free energy for each step of the reductive addition to the carbonyl carbon (red line) was observed. Experimentally, as shown in

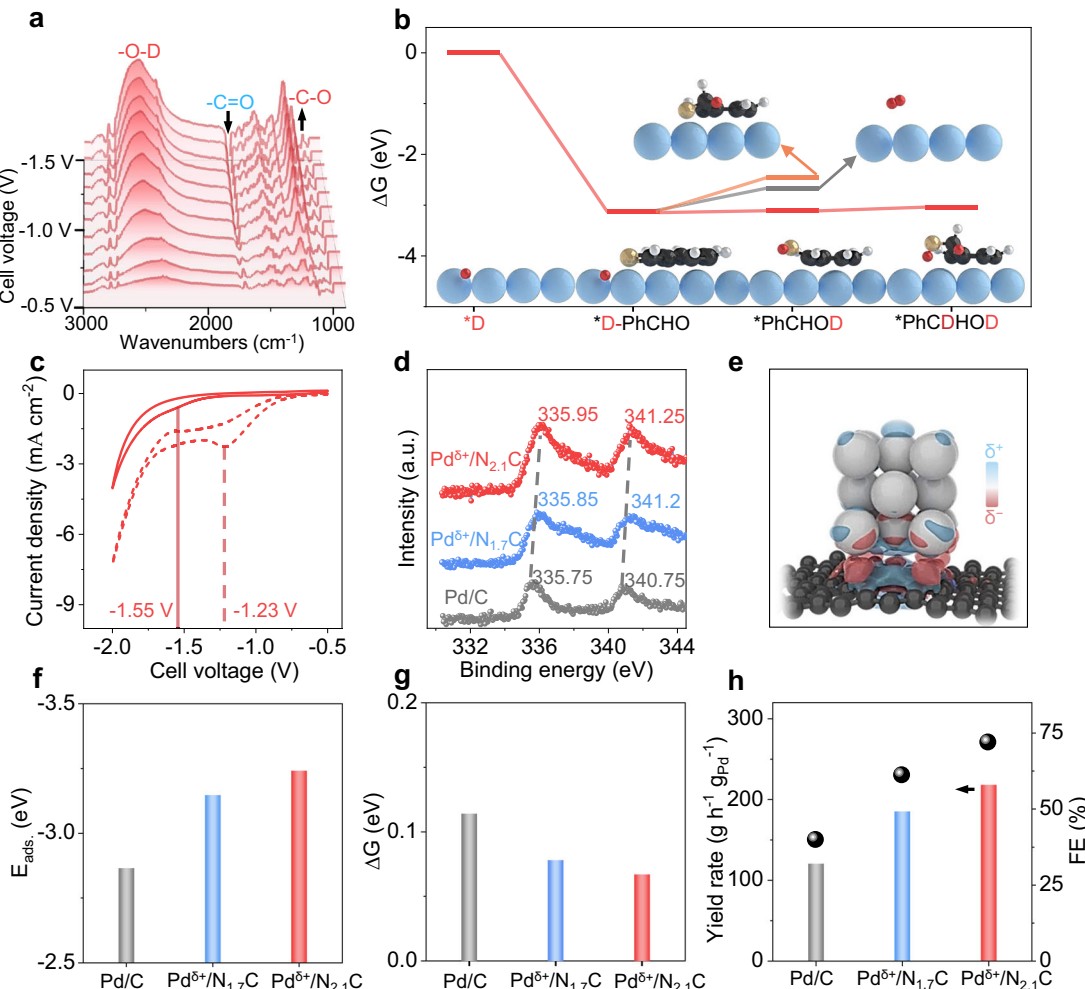

**Fig. 3 | Enhanced kinetics of deuterated reduction by an electron-deficient Pd cathode. a** In situ ATR-SEIRAS spectrum of the deuterated reduction reaction. **b** Gibbs free energy diagrams of each step of the deuterated reduction of benzaldehyde on the Pd$^{\delta+}$ surface and step-by-step adsorption configurations in the deuterated reduction of benzaldehyde to deuterated benzyl alcohol (blue, Pd; black, C; orange, O; red, D; white, H). **c** Cyclic voltammetry curves of Pd$^{\delta+}$/NC LDC with (solid line) and without (dashed line) benzaldehyde. **d** XPS spectra of Pd 3$d$ for Pd/C, Pd/N$_{1.7}$C and Pd/N$_{2.1}$C. **e** Difference in the charge density of Pd/N$_{2.1}$C; δ+: electron-deficient area, and δ−: electron-rich area (black, C; blue: N, Gray, Pd). **f** Calculated adsorption energies of benzaldehyde molecules on the Pd/C, Pd/N$_{1.7}$C and Pd/N$_{2.1}$C models. **g** Calculated Gibbs free energy of the rate-determining step of deuterated reduction on Pd/C, Pd/N$_{1.7}$C and Pd/N$_{2.1}$C. **h** Yield and FEs of deuterated benzyl alcohol of Pd/C LDC, Pd$^{\delta+}$/N$_{1.7}$C LDC and Pd$^{\delta+}$/N$_{2.1}$C LDC. Source data for Fig. 3a–h are provided as a Source Data file.

the cyclic voltammetry (CV) curves (Fig. 3c), after the addition of benzaldehyde, the −1.23 V peak (dashed line) disappeared, and a new peak at −1.55 V for the reduction of benzaldehyde appeared[32] (solid line); these results suggested the preferred pathway of the D$^+$ addition to benzaldehyde rather than the D$_2$ gas release. The second hydrogenation step, with an energy barrier of 0.067 eV, was the rate-determining step for the deuterated reduction of benzaldehyde; this value was much lower than that of the deuterium evolution reaction and could be used to explain the satisfactory FE for deuterated benzyl alcohol production over the Pd$^{\delta+}$/NC LDC.

Clearly, the electron density of the Pd nanocrystal in the Pd$^{\delta+}$/NC LDC dominates the deuteration progress. We gradually enhance the electron deficiency of the Pd nanocrystals from the neutral Pd of Pd/C by introducing more nitrogen dopants to the carbon sphere surface (Supplementary Data 1–4). The electronic structure of the carbon framework is modulated by incorporating nitrogen atoms, and this incorporation lowers the valence band and increases the conduction band[33,34]. The electronic characteristics of the composite are primarily determined by the work functions of the two components[35,36]. Nitrogen-rich carbon, with a relatively higher flat band potential than metallic Pd, accepts electrons from the Pd nanocrystals until their Fermi levels equilibrate (Fig. 3e and Figs. S15, 16). Since the work function of Pd is similar to that of nitrogen-doped carbon, the electron flow is highly sensitive to the nitrogen content. As the nitrogen content increases from Pd/C to Pd$^{\delta+}$/N$_{1.7}$C and Pd$^{\delta+}$/N$_{2.1}$C, the typical XPS peak of Pd gradually shifts to higher binding energy values as follows: from 335.75 eV to 335.95 eV for the Pd 3d$_{5/2}$ peak and from 340.75 eV to 341.25 eV for the Pd 3 d$_{3/2}$ peak (Fig. 3d and Fig. S17).

Due to the variation in the electron density of the Pd nanocrystals, the Pd/N$_{2.1}$C catalyst is the most electron-deficient Pd nanocrystals, exhibits the highest adsorption energy towards benzaldehyde (Fig. 3f) and promotes the rate-determining step by lowering its activation energy (Fig. 3g). This mechanism enables Pd$^{\delta+}$/NC LDC to exhibit optimal catalytic performance, and a Faradaic efficiency (FE) of 72% is attained (Fig. 3h).

## Electrocatalytic performance of reductive deuteration

The activation of D$^+$ over Pd$^{\delta+}$/NC LDC is universal for the solvent-free deuteration of a wide range of unsaturated substrates, including aldehydes, ketones, imines and alkenes (Fig. 4a and Table S3–6). The deuteration of benzaldehyde on the Pd/NC

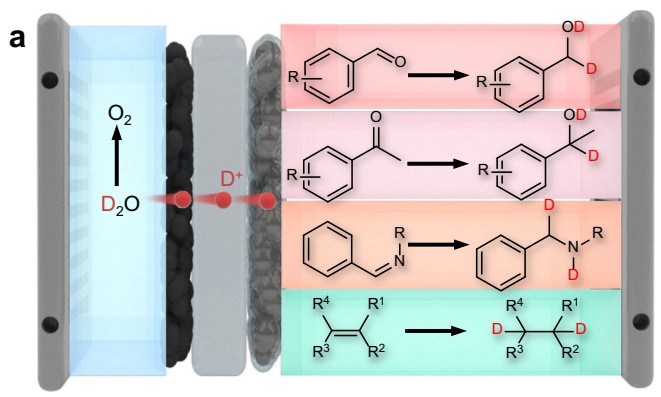

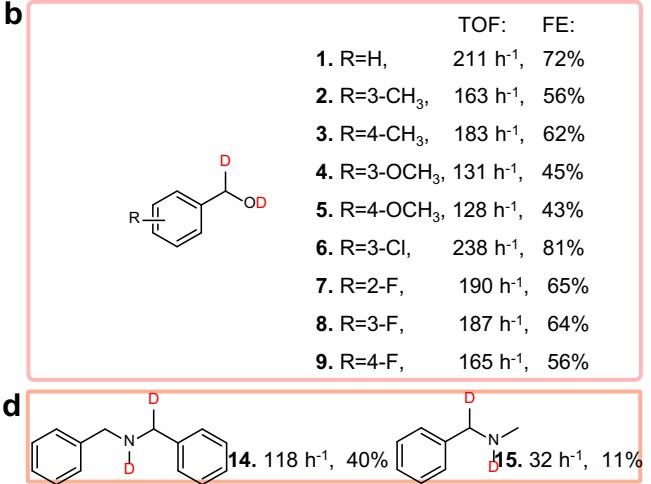

**Fig. 4 | Scope for the reductive deuteration of unsaturated bonds by direct D⁺ transfer over Pd^δ+/NC LDC. a** Schematic illustration of the deuterated reduction of aldehydes, ketones, imines, and olefins. **b** TOF and FEs of the reductive deuteration of aldehydes. **c** TOF and FE of the reductive deuteration of ketones. **d** TOF and FEs of the reductive deuteration of imines. **e** TOF and FEs of the reductive deuteration of aromatic olefins and aliphatic olefins. Source data for Fig. 4b–e are provided as a Source Data file.

cathode achieved a turnover frequency (TOF) of 211 h⁻¹; this value was 20 times greater than those of hydrogen isotope exchange (HIE) and the reductive deuteration methods (Fig. 4b)[37,38]. Additionally, ketones with a carbonyl group were efficiently deuterated (Fig. 4c) and achieved a turnover frequency (TOF) that was 2 to 6 times higher than those reported for the recent photocatalytic deuteration of ketones[39,40].

Nitrogen-containing compounds are widely present in organisms and pharmaceutical molecules and are highly important for their deuteration. Although the Faradaic efficiency for deuterated imines was relatively low (11–40%), these values still exceeded recent literature reports, whereas the turnover frequency (TOF), which benefits from the solvent-free reaction strategy, was over 10 times higher[18].

In addition, the electron-deficient Pd-based catalyst could also reduce alkenes to deuterated compounds (Fig. 4e); these compounds included aromatic alkenes (16–19) and aliphatic alkenes (20, 21), with a Faradaic efficiency exceeding 50%. In particular, the successful deuteration of 17 demonstrated the potential of this work for deuterated drugs since this compound is a precursor of ibuprofen[41].

### High-throughput deuteration

After optimizing the cathode and investigating the mechanisms, Pd^δ+/NC LDC is integrated into a commercially available PEM electrolyzer by using commercial MEA and GDL electrodes (Fig. 5a and Fig. S18), which realizes the deuterated benzaldehyde reaction at currents ranging from 0.02 A to 1 A (Fig. 5b and Fig. S19). At a current

of 1 A, the generation of D₂ gas at the cathode becomes significantly more pronounced, hindering the diffusion of the substrate to the electrode surface with a relatively lower reaction rate. This device is capable of continuously producing deuterated benzyl alcohol for more than 500 h under 0.02 A current, resulting in 13 g of product (Fig. 5c) with excellent stability of Pd^δ+/NC LDC (Figs. S20–23). Moreover, the GC–MS spectrum of the anode product revealed that only trace amounts of benzaldehyde (<0.05%, 18 h) passed through the MEA membrane, and no reduction products were detected (Fig. 5d). A test was subsequently conducted at 0.6 A (Fig. 5e). This non-contact reduction strategy significantly facilitates the separation and reuse of the deuterium source from the electrolyzer. The majority of D₂O at the anode, noncontact with the substrate, can be reused directly for following runs of reactions (Fig. S18). A small part of the D₂O, crossed the proton exchange membrane after the reaction at 600 mA for 20 h. Owing to the solvent-free reaction strategy, the poor miscibility between deuterated water and organic compounds, a barrier for one-pot deuteration reactions, enables easy separation of the organic compounds and permeated deuterated water with a recovery rate above 98.8% simply by pipetting (Fig. 5f and Fig. S24). Replacing new commercial IrO₂ MEA with the same Pd/NC liquid diffusion cathode could maintain the performance of Pd^δ+/NC-based electrolyzer under high current experimental conditions (600 mA), indicating the possible effect of the deactivated IrO₂ MEA on the cell voltage increase (Fig. S25). Compared with other countless approaches, such as Pd membrane reactors[18,42–47], this device has the ability to operate at a product cost of 0.25 CNY/mmol

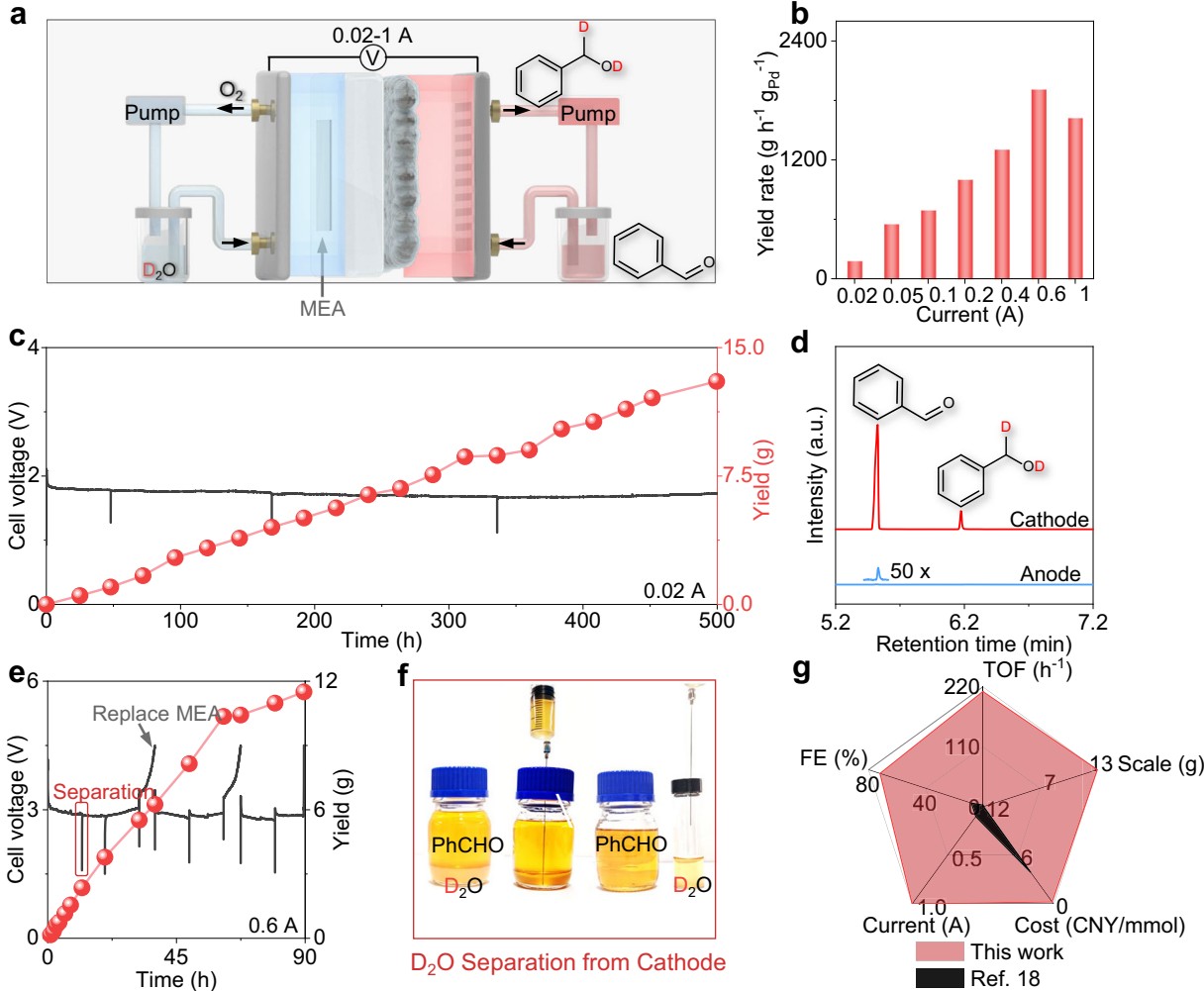

**Fig. 5 | Integration of the Pd^δ+/NC LDC into the membrane electrode assembly (MEA) electrolyzer. a** Schematic diagram of the MEA electrolyzer device. **b** Yield of the device at different currents. **c** Chronopotentiometry curve and yield of the reductive deuteration of an aldehyde at 0.02 A for 500 h. **d** GC–MS spectra of the cathode and anode solutions during the reaction process. **e** Chronopotentiometry curve and yield of the reductive deuteration of an aldehyde at 0.6 A. **f** Simple and efficient separation of permeated D₂O in the cathode solution. **g** Advantages of this work over other approaches[18]. Source data for Fig. 5b−e and Fig. 5g are provided as a Source Data file.

($D_2O$ contributed 99.7% and Pd/NC cathode contributed 0.3%), with a current density ranging from 0.02 A to 1 A, reaching a Faradaic efficiency of over 70%, while simultaneously producing ten-gram-scale products. These results confirm the economic feasibility of the Pd/NC cathode for industrial applications.

## Discussion

In this study, a deuterium ion diffusion-based all-solid electrolyzer is designed for efficient, solvent-free deuteration with separated substrates and deuterium sources. Deuterium ions generated at the anode are transported to the cathode, where electron-deficient Pd facilitates their addition to the unsaturated compounds and the reaction energy barrier is lowered. The Pd/NC cathode achieves a Faradaic efficiency (FE) of 72% and enables ten-gram-scale production of deuterated benzyl alcohol. Additionally, using $D_2O$ as the deuterium source, the Pd/NC cathode supports the formation of the C(sp³)−D bonds from aldehydes (FEs: 43−72%, Selectivity: 88−99%), ketones (FEs: 31−90%, Selectivity: 99%), imines (FEs: 11−40%, Selectivity: 99%) and alkenes (FEs: 55−97%, Selectivity: 99%). The Pd-based electrocatalyst device has great promise for large-scale, cost-effective deuteration, with potential applications in manufacturing.

## Methods

### Chemicals and materials

Ketjen Black (ECP-600JD), Nitric acid ($HNO_3$, AR, 65.0%), Dicyandiamide (DCDA, 99.5%), Palladium (II) Chloride ($PdCl_2$, 98%), benzaldehyde ($C_7H_6O$, 99.0%), Ruthenium (IV) Oxide ($RuO_2$, Ru: 59.5%+), Sodium Borohydride ($NaBH_4$, 98%) was purchased from Aladdin. Nafion solution (10%) and proton exchange membrane (PEM) were purchased from DuPont. Carbon cloth (W0S1009) was purchased from CeTech and cut into small pieces ($2 \times 2$ cm²).

### Preparation of $N_XC$

Commercially available Ketjen Black (ECP-600JD, KB) was used as the starting carbon material. Specifically, 2 g of KB was put inside a three-neck flask containing 230 mL 65% $HNO_3$ solution and 70 mL DI water. The whole device was connected with a reflux system afterwards under well-stirred conditions, and the temperature was fixed at 80 °C for 24 h. The black called OC was obtained by centrifuging and washing. The mixture of 1 g of OC and 5 g/10 g of dicyandiamide was heated at 800 °C for 2 h with a heating rate of 2.5 °C/min under nitrogen atmosphere. The as-obtained black powders after cooling down to room temperature were named $N_XC$ (X represents the content of N element, determined by XPS (Table S1)).

## Preparation of Pd/N$_X$C

Typically, N$_X$C (50 mg), PdCl$_2$ (5 wt% Pd), were uniformly dispersed into 30 ml deionized water by ultrasonic and vigorous stirring. After stirring for 12 h, the pH value of the above solution was adjusted to >10 with 2 moL L$^{-1}$ NaOH. After that, 10 mL of NaBH$_4$ solution (molar ratio of NaBH$_4$ to metal = 40:1) was slowly added into the above suspension under stirred for 4 h. The Pd N$_X$C was obtained by centrifugated, washed with deionized water and ethanol and dried at 60 °C overnight. Pd/C was obtained in the same method, except that N$_X$C was replaced by Ketjen Black.

## Characterizations

X-ray diffraction (XRD) patterns were carried out on a Bruker D8 Advance X-ray diffractometer with a Cu Kα radiation source (λ = 1.5418 Å). High-resolution transmission electron microscopy (HRTEM) images were performed on a JEM-2100F microscope with an acceleration voltage of 200 kV. Scanning electron microscopy (SEM) was measured on an FEI Nova NanoSEM 450 field emission scanning electron microscope. The inductively coupled plasma atomic emission spectroscopy (ICP-AES) measurements were conducted on an iCAP6300 spectrometer for palladium and copper element concentrations with a detection limit of 1 ug/L. X-ray photoelectron spectroscopy (XPS) was conducted on a Thermo ScientificTM K-AlphaTM+ spectrometer equipped with a monochromatic Al Kα X-ray source (1486.6 eV). All the peaks were calibrated by the binding energy of 284.8 eV of the C 1s spectrum.

## Construction of the test device

**Anode electrode.** 3.5 mg RuO$_2$, 700 μL EtOH, 200 μL H$_2$O and 100 μL Nafion solution were mixed to form a homogeneous ink by sonicated for 2 h. 1.2 mL of ink was dropped on the titanium mesh evenly at certain area (2 × 2 cm$^2$). The obtained electrode was heated at 120 °C for 1 h.

**Pd$^{δ+}$/NC LDC.** 5 mg catalysts, 700 μL EtOH, 200 μL H$_2$O and 100 μL Nafion solution were mixed to form a homogeneous ink by sonicated for 0.5 h. 0.8 mL of ink was dropped on the carbon cloth evenly at certain area (2 × 2 cm$^2$). The catalyst loading is 1 mg cm$^{-1}$. The obtained electrode was heated at 120 °C for 1 h.

**Preparation of membrane.** N115 (4 cm × 4 cm) with a thickness of 127 microns and were used as proton exchange membrane. It was pre-treated before use. The pre-treatment involved treating the PEM with 5 wt% hydrogen peroxide at 80 °C for 1 h, followed by soaking in deionized water for 0.5 h. This was then followed by treatment with 5 wt% sulfuric acid at 80 °C for 1 h, again followed by soaking in deionized water for 0.5 h. Finally, the PEM was soaked in deuterated water to replace the hydrogen, and stored in deuterated water.

**Reduction of benzaldehyde to deuterated benzyl alcohol.** All of the electrochemical experiments were performed in a two-electrode system on an electrochemical station (CT-4000, Neware Technology Limited). The anode side was circulated with 2 mL pure D$_2$O at 1 mL min$^{-1}$. The cathode side was circulated with 2 mL pure benzaldehyde at 1 mL min$^{-1}$. A rubber of 0.5 mm thickness was used as a gasket. The generated cathode products were analyzed by GC-MS to determine the components of products and calculate the conversion and selectivity. All the electrochemical experiments were carried out at room temperature (25 °C).

The Faradaic efficiency (FE) is calculated as follows:

$$FE\,(\%) = \frac{n \times Ni \times F}{Q} \times 100\%$$

where $Ni$ (mole) is the number of moles for the specific product; $n$ is the number of electrons exchanged for product formation, which is 2e

in this reaction; $F$ is the Faraday constant of 96487 C mol$^{-1}$; $Q$ is the passed charge.

**Electrochemical impedance spectroscopy (EIS) and distribution of relaxation times (DRT) analysis.** Potentiostatic EIS measurements are carried out in the range from 0.1 Hz to 1 MHz and an amplitude of 50 mV. The distribution of relaxation times (DRT) is used to analyze the EIS data, respectively

DRT impedance, $Z_{DRT}(f)$, at a frequency $f$, can be expressed as[24–26]

$$Z_{DRT} = R_\infty + \int_{-\infty}^{\infty} \frac{\gamma(\ln\tau)}{1 + i2\pi f\tau}\, d\ln\tau \tag{1}$$

where, $R\infty$, $\tau$, $\gamma(\ln\tau)$ and $f$ are an ohmic resistance, relaxation time, distribution function of relaxation times, and frequency. Moreover, polarization resistance, $R_{pol}$, is computed by

$$R_{pol} = \int_{-\infty}^{\infty} \gamma(\ln\tau) d\ln\tau \tag{2}$$

The DRT analysis is performed with a regularization parameter of 0.001 as previously reported[24,26].

**Cost of deuteration process.** The cost of the deuterium reagent for palladium membrane deuteration was obtained from the literature as 0.56 USD/mmol (about 3.92 CNY/mmol)[18]. The cost of this work was calculated by determination of the cathode and the amount of D$_2$O consumed per mol product. The prices of D$_2$O from Adamas is 176 CNY/mol.

For 1 h deuterated reduction of benzaldehyde to benzyl alcohol at 20 mA:

$$Cost\,of\,D_2O = \frac{charge\,passed\,through\,for\,1\,h\,at\,20\,mA}{Faraday's\,constant \times 2}$$
$$\times price\,of\,[D_2O]_{per\,mol}$$

$$= \frac{0.02\,A \times 3600\,s}{96485\,C/mol \times 2} \times 176\,CNY/mol$$

$$= 0.0657\,CNY$$

$$Cost\,of\,per\,mol\,product = \frac{Cost\,of\,D2O}{mols\,of\,product}$$

$$= \frac{0.0657\,CNY}{268.4 \times 10^{-6}\,mol}$$

$$= 0.25\,CNY/mmol$$

For the contribution of Pd/NC cathode cost:

$$Contribution\,of\,Pd\,NC\,cathode\,cost$$
$$= \frac{cost\,of\,Pd/NC\,cathode}{cost\,of\,Pd/NC\,cathode + cost\,of\,D_2O}$$

$$= 0.3\%$$

**Theoretical calculations.** Theoretical calculations were conducted using the Vienna Ab Initio Simulation Package (VASP), employing the generalized gradient approximation (GGA) with the Perdew–Burke–Ernzerhof (PBE) exchange-correlation functional[48,49]. A plane-wave energy cutoff of 450 eV, an energy convergence criterion of 10$^{-5}$ eV, and a residual force of 0.02 eV/Å were applied. K-points were

set to a 3 × 3 × 1 grid for both geometry optimization and electronic structure analysis. The van der Waals dispersion correction was incorporated using the DFT-D3 method. Spin polarization was included when necessary.

For the Bader charge analysis of the Pd metal and NC support, an N-doped graphene support (5 × 5) was modeled based on the C/N ratio and the concentration of pyrrole/pyridinic N dopants derived from XPS data (Fig. S17 and Table S1). The Pd cluster was positioned above the N-doped graphene support (Figs. S15–16 and Supplementary Data 1–2), with a vacuum layer of 20 Å.

To model the electron-deficient Pd system, we assigned a charge of −0.011 e⁻ and −0.009 e⁻ per Pd atom to create the $Pd^{\delta+}/N_{2.1}C$ and $Pd^{\delta+}/N_{1.7}C$ model, allowing us to study the interaction between the reactant and the catalyst surface based on the Bader charge analysis (Supplementary Data 3–4). The crystal unit model with the selected (111) facet was used to construct both the neutral and electron-rich Pd slab models, guided by HETEM and XRD data. The Gibbs free energy change (ΔG) for each reaction step was computed using the following equation ($T$ = 298.15 K):

$$\Delta G = \Delta E + \Delta ZPE - T\Delta S$$

where $\Delta E$, $\Delta ZPE$, and $\Delta S$ represent the energy differences in the reaction energy, zero-point energy, and entropy, respectively.

## Data availability
All the relevant data are included in this paper and its Supplementary Information. The data that support the findings of this study are available from the corresponding authors upon request. Source data are provided with this paper.

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

## Acknowledgements

This work was supported by the National Natural Science Foundation of China (22071146 for X.-H.L. and 21931005 for J.-S.C.), Shanghai Science and Technology Committee (23XD1421800 for X.-H.L.), Shanghai Shuguang Program (21SG12 for X.-H.L.), and Shanghai Municipal Science and Technology Major Project.

## Author contributions

X.-H.L. and X.-F.Z. designed the experiments. X.-F.Z. planned and performed catalyst synthesis, conducted the performance tests and analyzed data. S.-N. Z. finished the theoretical calculation. Z.Z. contributed to the design and assembly of the performance testing device. Z.Z. and B.-L.L. helped to the design and assembly of the in situ ATR-SEIRAS. K.-Y.L. contributed to the substrate scope section. X.-H.L. and X.-F.Z. cowrote the original manuscript. X.-H.L. and J.-S.C. oversaw all of the research phases. All of the authors discussed the results and commented on the paper.

## Competing interests

The authors declare no competing interests.
