## [Transparent Peer Review file · Nature Communications]

Boosted high-throughput D⁺ transfer from D₂O to unsaturated bonds via Pd^{δ+} cathode for solvent-free deuteration

Corresponding Author: Professor Xin-Hao Li

Version 0:

Reviewer comments:

Reviewer #1

(Remarks to the Author)

This manuscript reported a novel synthesis strategy of deuterated organic compounds. This work achieved a solvent-free, green and sustainable deuterium reduction synthesis by designing a two-electrode reactor with deuterium water as the deuterium source at the anode coupled with deuterium reduction of benzaldehyde at the cathode. Particularly, the cathode catalyst Pd/NC LDC with tunable electronic structure achieves efficient deuterium substitution selectivity and Faraday efficiency. In addition, stable gram-scale synthesis over a long period of time was achieved by optimizing the process conditions of the reactor. This work has attractive potential for commercial practical applications. Therefore, I suggest that the manuscript may be accepted after the following issues have been well-resolved.

1. In Figure 1e, the NC has no metal active center but has some catalytic performance, please explain it.
2. In Figure 1g, the Faradaic efficiency at 20 mA current is significantly higher than the others, so please make the necessary explanations.
3. In Figures 2b and S6, the $\ln(t)$ is missing units, please add. Meanwhile, Nyquist plots should be square, with x- and y-axes having the same length scale to identify deviations from ideal behavior.
4. In Table S3, the selectivity of entry 6 is 88%, it is better to give other products and to give a sample explanation.
5. The concern in the synthesis of deuterated molecules is the deuterium incorporation. It is desirable for the authors to give the deuterium incorporation of deuterated benzyl alcohol.
6. Some experimental details have to be given, such as the parameters of the PEM.
7. The authors are requested to carefully correct the formatting of this manuscript, e.g., some unit inconsistencies (mg/cm²), redundant spaces (C 1s), and singular and plural (spectrum, FE).

Reviewer #2

(Remarks to the Author)

The manuscript entitled "Boosted high-throughput D⁺ transfer from pure D₂O to unsaturated bonds via a Pd^{δ+} cathode for solvent-free deuteration" has been reviewed. The authors present a deuterium ion diffusion-based electrolyzer for high-throughput, solvent-free electrochemical deuteration by using a Pd/N-doped carbon liquid diffusion cathode (Pd^{δ+}/NC LDC) with tunable electron deficiencies to enhance selective deuteration. This design enables highly efficient deuteration of various unsaturated compounds, including aldehydes, ketones, imines, and alkenes and over 99% selectivity and 72% Faradaic efficiency for deuterated benzyl alcohol. DRT and DFT calculations confirm that the Pd^{δ+}/NC LDC enhances reaction performance by reducing the energy barrier of the rate-determining step. With the implementation of commercial MEAs and GDLs, Ampere-level current density deuteration of benzaldehyde was achieved, enabling ten-gram-scale production of deuterated benzyl alcohol. This work offers a promising strategy for the scalable synthesis of deuterated organic compounds which could benefit pharmaceutical drug development and industrial isotope labeling. Accordingly, the manuscript is recommended to be published in Nature Communications after minor corrections. The authors should address the following points.

1. The authors achieved benzaldehyde deuteration reduction in the current range of 0.02 A to 1 A. However, Figure 5b shows that the reaction rate at 1 A is lower than that at 0.6 A. It is better to provide an explanation for this phenomenon.
2. The authors compared the reaction performance of different catalysts (Pd/C, Pd/N1.7C and Pd/N2.1C) and attributed the

difference to their ability to lower the energy of the rate-determining step. Although this is presented in Figure 3g and Figure S12, it is helpful to give the specific values in supplementary information.

3. The authors have only provided the Bader charge analysis for Pd/2.1C, and the Bader charge analysis for Pd/N1.7C should also be given.

4. In Figure 5e, the cell voltage of the device increases at approximately 30 hours and 60 hours, and after the replacement of MEA, the device can continue to operate. The authors are advised to provide a more detailed explanation.

Reviewer #3

(Remarks to the Author)

This paper presents an interesting approach to solvent-free deuteration using a Pd^{δ+}/NC-based electrolyser; however, it exhibits several significant drawbacks that limit its applicability and scalability. The substrate scope is narrow, and the variable Faradaic efficiency across different substrates limits the method's reliability. Additionally, the reliance on Pd^{δ+}/NC cathodes introduces potential issues with cost and material availability, which could hinder large-scale adoption. While the recycling of D₂O is mentioned, it is not sufficiently explored in terms of practical implementation. Moreover, the study does not provide enough detail on potential side reactions or the long-term stability and durability of the Pd^{δ+}/NC electrocatalyst, both of which are critical factors for real-world applications (see the relevant literature: *J. Org. Chem.* 2024, 89, 10, 7065–7075; *Adv. Synth. Catal.* 2023, 365, 476–481). Given these limitations, the paper does not sufficiently address the challenges at hand and is not suitable for publication in *Nature Communications*.

Version 1:

Reviewer comments:

Reviewer #1

(Remarks to the Author)

The reviewer's concerns have been well-addressed. Publication of the revised version on *Nature Communications* could be now recommended.

Reviewer #2

(Remarks to the Author)

I appreciate the authors's efforts in addressing all reviewers' concerns. Upon reviewing the responses and the revised manuscript, I find that the manuscript has been significantly improved. I am very satisfied with the improvements and recommend its rapid publication.

Reviewer #3

(Remarks to the Author)

I am happy with the revision made. The manuscript can be accepted for publication.

Reviewer #1 (Remarks to the Author):

This manuscript reported a novel synthesis strategy of deuterated organic compounds. This work achieved a solvent-free, green and sustainable deuterium reduction synthesis by designing a two-electrode reactor with deuterium water as the deuterium source at the anode coupled with deuterium reduction of benzaldehyde at the cathode. Particularly, the cathode catalyst Pd/NC LDC with tunable electronic structure achieves efficient deuterium substitution selectivity and Faraday efficiency. In addition, stable gram-scale synthesis over a long period of time was achieved by optimizing the process conditions of the reactor. This work has attractive potential for commercial practical applications. Therefore, I suggest that the manuscript may be accepted after the following issues have been well-resolved.

C1: *In Figure 1e, the NC has no metal active center but has some catalytic performance, please explain it.*

R1: Thoughtful comments. The bare NC support with heteroatom dopants has an opened band gap and thus the defect sites for possible activation of substrates. The NC-based cathode exhibits a detectable conversion of 47 μmol benzaldehyde (Figure 1e) with a FE of 6%, which is only about one-tenth of Pd/NC cathode and one-fifth of Pd/C cathode. All these results demonstrated the role of Pd metal nanoparticles as highly active centers for reductive deuteration of benzaldehyde. We have added this explanation to the detailed discussion of Figure 1e in the revised manuscript.

C2: *In Figure 1g, the Faradaic efficiency at 20 mA current is significantly higher than the others, so please make the necessary explanations.*

R2: Thanks for the reviewer's suggestions. The Pd/NC cathode exhibited a wide electrochemical window for deuterated benzyl alcohol production, with FE values > 59% at different current densities ranging from 10 to 40 mA. Slightly increase in the current density from 10 to 20 mA leads to an elevated FE value from 59% to 72%. Further amplifying the current density to 40 mA may generate more hydrogen gas bubbles, resulting in a slight decrease in the FE values to 63-65% presumably due to the relatively sluggish substrate diffusion efficiency at the three-phase interface over the cathode. We have added this explanation to the detailed discussion of Figure 1g of the revised manuscript.

C3: *In Figures 2b and S6, the $\ln(t)$ is missing units, please add. Meanwhile, Nyquist plots should be square, with x- and y-axes having the same length scale to identify deviations from ideal behavior.*

R3: Thanks for the reviewer's kind suggestions. We have corrected the Figure 2b and Figure S6-S8, and also made revisions to the Nyquist plots.

C4: *In Table S3, the selectivity of entry 6 is 88%, it is better to give other products and to give a sample explanation.*

R4: We have added detailed information on the byproduct of entry 6 in Table S3 in the revised supplementary Information according to the reviewer's kind suggestion.

C5: *The concern in the synthesis of deuterated molecules is the deuterium incorporation. It is desirable for the authors to give the deuterium incorporation of deuterated benzyl alcohol.*

R5: Expert comments. We calculated the deuterium percentage of deuterated benzyl alcohol and added the detailed method and results to Figure S5 of the revised supplementary Information.

The NMR data demonstrated the formation of deuterated benzyl alcohol as the only detected product. Based on the analytic results in Figure S5, the spectrum of electrochemical reductive deuteration product, benzenemethan-d-ol-d, was analyzed as follows. The hydrogen peak at 4.50 ppm corresponded to hydrogen of -CHD- and was integrated into 0.99. This residual signal is derived from -CHO of benzaldehyde. The D-incorporation of this site is:

$$\begin{aligned}\text{Deuterium percentage} &= \left(\frac{\text{deuterium atom}}{\text{number of labelling sites}} \right) \times 100\% \\ &= \left(\frac{2 - 1.02}{1} \right) \times 100\% \\ &= 98\%\end{aligned}$$

C6: *Some experimental details have to be given, such as the parameters of the PEM.*

R6: We added the parameters of all subunits of the electrolyzer, including PEM, gas diffusion layer, current collector and gasket to the Methods of the revised manuscript and the figure caption of Figure S16 of the revised supporting information.

C7: *The authors are requested to carefully correct the formatting of this manuscript, e.g., some unit inconsistencies (mg/cm²), redundant spaces (C 1s), and singular and plural (spectrum, FE).*

R7: Thanks to the reviewer's help in polishing our paper. We have carefully removed the typos and errors in the revised manuscript and supporting information accordingly.

We highly appreciate the reviewer's kind suggestions and efforts to make our paper better.

Reviewer #2 (Remarks to the Author):

The manuscript entitled “Boosted high-throughput D⁺ transfer from pure D₂O to unsaturated bonds via a Pd^{δ+} cathode for solvent-free deuteration” has been reviewed. The authors present a deuterium ion diffusion-based electrolyzer for high-throughput, solvent-free electrochemical deuteration by using a Pd/N-doped carbon liquid diffusion cathode (Pd^{δ+}/NC LDC) with tunable electron deficiencies to enhance selective deuteration. This design enables highly efficient deuteration of various unsaturated compounds, including aldehydes, ketones, imines, and alkenes and over 99% selectivity and 72% Faradaic efficiency for deuterated benzyl alcohol. DRT and DFT calculations confirm that the Pd^{δ+}/NC LDC enhances reaction performance by reducing the energy barrier of the rate-determining step. With the implementation of commercial MEAs and GDLs, Ampere-level current density deuteration of benzaldehyde was achieved, enabling ten-gram-scale production of deuterated benzyl alcohol. This work offers a promising strategy for the scalable synthesis of deuterated organic compounds which could benefit pharmaceutical drug development and industrial isotope labeling. Accordingly, the manuscript is recommended to be published in Nature Communications after minor corrections. The authors should address the following points.

C1: *The authors achieved benzaldehyde deuteration reduction in the current range of 0.02 A to 1 A. However, Figure 5b shows that the reaction rate at 1 A is lower than that at 0.6 A. It is better to provide an explanation for this phenomenon.*

R1: Expert comments. After optimizing the cathode and investigating the mechanisms, Pd/NC cathode is integrated into a commercially available PEM electrolyzer by using commercial MEA and GDL electrodes (Figure 5a and Figure S16), which realizes the deuterated benzaldehyde reaction at currents ranging from 0.02 A to 1 A (Figure 5b and Figure S17). At a current of 1 A, the generation of D₂ gas at the cathode becomes significantly more pronounced, hindering the diffusion of the substrate to the electrode surface with a relatively lower reaction rate. We have added this explanation to the detailed discussion of Figure 5b of revised manuscript.

C2: *The authors compared the reaction performance of different catalysts (Pd/C, Pd/N1.7C and Pd/N2.1C) and attributed the difference to their ability to lower the energy of the rate-determining step. Although this is presented in Figure 3g and Figure S12, it is helpful to give the specific values in supplementary information.*

R2: According to the reviewer's kind reminder, we added the specific values for the free energy changes at each step of the reaction process to the Figure S12 of revised supplementary Information.

C3: *The authors have only provided the Bader charge analysis for Pd/2.1C, and the Bader charge analysis for Pd/N1.7C should also be given.*

R3: Kind suggestion. Bader charge analysis for Pd/N_{1.7}C was also added to the Figure S14 of revised supplementary information.

C4: *In Figure 5e, the cell voltage of the device increases at approximately 30 hours and 60 hours, and after the replacement of MEA, the device can continue to operate. The authors are advised to provide a more detailed explanation.*

R4: Expert comments. Replacing new commercial IrO₂ MEA with the same Pd/NC liquid diffusion cathode could maintain the performance of Pd^{δ+}/NC-based electrolyser under high current experimental conditions (600 mA), indicating the possible effect of the deactivated IrO₂ MEA on the cell voltage increase. Indeed, detailed XRD analysis results of the used IrO₂ MEA reveals an obvious deconstruction of the IrO₂ materials after a 32-hour uses. We also added the XRD patterns (Figure S23) of the MEA before and after uses for 32 h at 0.6 A and detailed explanation to the revised supplementary Information.

We highly appreciate the reviewer's kind suggestions and efforts to polish our paper.

Reviewer #3 (Remarks to the Author):

This paper presents an interesting approach to solvent-free deuteration using a Pd^{δ+}/NC-based electrolyser; however, it exhibits several significant drawbacks that limit its applicability and scalability. The substrate scope is narrow, and the variable Faradaic efficiency across different substrates limits the method's reliability. Additionally, the reliance on Pd^{δ+}/NC cathodes introduces potential issues with cost and material availability, which could hinder large-scale adoption. While the recycling of D₂O is mentioned, it is not sufficiently explored in terms of practical implementation. Moreover, the study does not provide enough detail on potential side reactions or the long-term stability and durability of the Pd^{δ+}/NC electrocatalyst, both of which are critical factors for real-world applications (see the relevant literature: J. Org. Chem. 2024, 89, 10, 7065–7075; Adv. Synth. Catal. 2023, 365, 476–481). Given these limitations, the paper does not sufficiently address the challenges at hand and is not suitable for publication in Nature Communications.

Reply: We thank the reviewer for the constructive suggestions, and we added more data and detailed discussion to clarify the concerns by reviewers with detailed point-to-point responses as following:

C1: *The substrate scope is narrow, and the variable Faradaic efficiency across different substrates limits the method's reliability.*

R1: Kind suggestions. We have tried to test the possibility of our method for the reductive deuteration of all common C=X (X=O, N, C) based substrates, including but not limited to aldehydes, ketones, imines, and alkenes, with more than 21 representative substrates (for experimental detail please see Table S3-6 of the revised supporting information). It should be noted that all substrates are tested under fixed conditions with acceptable FE values between 11-98%. For better comparison, we could optimize the FE for deuteration of benzaldehyde to a record-high 72% via the Pd/NC cathode, which is 10 times of that of the state-of-the-art device (Pd membrane reactor) for the same reaction. More importantly, it is the first report of achieving ten-gram-scale production of deuterated benzyl alcohol at an Ampere-level current density for over 90 hours using pure deuterated water and substrates. Our FE data also far surpass the reported values of bench-marked contactless reactor for deuterium transfer of aldehydes, imines, and alkenes, respectively, as updated in Table S3-6 of the revised supplementary Information. The Pd^{δ+}/NC based electrolyzer has shown great promise for large-scale and cost-effective deuteration, with potential applications in mass production of various deuterated alcohols, deuterated amines and deuterated alkanes. We are also open to make further revisions upon additional comments from the reviewers in the future.

C2: *Additionally, the reliance on Pd^{δ+}/NC cathodes introduces potential issues with cost and material availability, which could hinder large-scale adoption.*

R2: To clarify the reviewer's concern on the cost and material availability, we further calculated the overall cost of the electrode for reductive deuteration of benzaldehyde. The catalyst loading on a cathode is 4 mg (1 mg cm⁻²), with a Pd content of 3 wt% based on ICP results (Table S2). Meanwhile, in the long-cycle experiment, such a cathode was used

to conduct a 500-hour reaction at 20 mA (Figure 5c), yielding approximately 118 mmol of the deuterated benzyl alcohol by using 180 mmol of D₂O. The cost of Pd/NC cathode per mmol product is 0.3% of the total cost of the deuteration process, while the consumption of D₂O contribute 99.7% of the total cost.

$$\begin{aligned} \text{Contribution of Pd/NC cathode cost} &= \frac{\text{cost of Pd/NC cathode}}{\text{cost of Pd/NC cathode} + \text{cost of D}_2\text{O}} \\ &= 0.3\% \end{aligned}$$

As a result, we focus on the design of the process that does not depend on the direct contact of the deuterium source with the substrate to recycle the deuterium source and further depress cost of in this work. We added this discussion to conclusion and revised supplementary Information.

C3: *While the recycling of D₂O is mentioned, it is not sufficiently explored in terms of practical implementation.*

R3: Expert comments. We explored the separation of deuterium water from the substrate with the separation operation and related pictures have been added to the revised manuscript (Figure 5d-f). This study achieves the solvent-free deuteration reduction of pure substrate at Ampere-level with record-high FE values, benefiting the reduced consumption and separation cost of deuterium source for practical implementation. This non-contact reduction strategy significantly facilitates the separation and reuse of the deuterium source from the electrolyzer. The majority of D₂O at the anode, noncontact with the substrate, can be reused directly for following runs of reactions (Figure S16). A small part of the D₂O, crossed the proton exchange membrane after the reaction at 600 mA for 20 h, can be collected with a recovery rate above 98.8% simply by pipetting (Figure S22). We also added detailed data (Figure S16 and S22) and discussion on this point in the revised version.

C4: *Moreover, the study does not provide enough detail on potential side reactions or the long-term stability and durability of the Pd^{δ+}/NC electrocatalyst, both of which are critical factors for real-world applications (see the relevant literature: J. Org. Chem. 2024, 89, 10, 7065–7075; Adv. Synth. Catal. 2023, 365, 476–481).*

R4: Thoughtful comments. We have added the detailed information on the byproducts over the cathode in Table S3 in revised supplementary Information. We also agree with the reviewer that long-term stability and durability of the cathode is important for real uses. We carefully read the suggested literature and cited them in the revised version as Ref. 11-12. As compared with the reported data (5 h for reductive deuteration of aldehyde/ketone at 30 mA in mixed electrolyte) in Ref. 11-12, the Pd/NC liquid diffusion cathode based electrolyzer can achieve more than 500 hours of operation at 20 mA, and more than 90 hours of operation at 600 mA, again speaking for the advantages of our current method on both of the current density (more precisely the product rate) and durability for reductive deuteration reaction. We added the XRD (Figure S20) and XPS (Figure S21) analysis results of the Pd/NC liquid diffusion cathode before and after the 500-h reaction, which indicated the stability of the catalyst on the electrode surface in the revised supplementary Information. We also added detailed discussion on this point in the revised manuscript.

Once again, we highly appreciate the reviewer's valuable suggestions and efforts to refine our paper.